# The Maximal Urethral Pressure at Rest and during Normal Bladder Filling Is Only Determined by the Activity of the Urethral Smooth Musculature in the Female

**DOI:** 10.3390/jcm12072575

**Published:** 2023-03-29

**Authors:** Pieter L. Venema, Guus Kramer, Gommert A. van Koeveringe, John P. F. A. Heesakkers

**Affiliations:** Department of Urology, Maastricht University Medical Center, P. Debeyelaan 25, 6229 HX Maastricht, The Netherlands

**Keywords:** external urethral sphincter, urethral anatomy, urethral physiology, urethral pressure profile, stress incontinence, lower urinary tract symptoms

## Abstract

The aim of this opinion paper is to determine the entities that define the maximal urethral pressure (MUP) during rest and during bladder filling that is needed to guarantee continence in females. For the development of this opinion, the literature was searched for via the Pubmed database and historic sources. Animal studies indicate that the maximal urethral pressure is determined by the smooth muscle activity in the mid-urethra. Additionally, during increased smooth muscle tone development, the largest sympathetic responses are found in the middle part of the urethra. This could be confirmed in human studies that are unable to find striated EMG activity in this area. Moreover, the external urethral striated sphincter is situated at the distal urethra, which is not the area with the highest pressure. The external urethral sphincter only provides additional urethral pressure in situations of exertion and physical activity. From a physics point of view, the phasic pressure of the external striated sphincter at the distal urethra cannot be added to the tonic pressure generated by the smooth muscle in the mid-urethra. The assertion that mid-urethral pressure is the result of different pressure forces around the urethra, including that of the external striated sphincter, is not supported by basic research evidence combined with physical calculation and should therefore be considered a misconception in the field of functional urology.

## 1. Introduction

The literature claims that urethral pressure is determined by multiple factors. In this paper, we argue that the urethral pressure reflects urethral smooth muscle activity only. An amply-referenced article by Rud et al. [1] states that, at rest, the striated muscles of the urethra contribute to one-third of the pressure of the urethra, and both the smooth muscles and vascular bed each add up an extra one-third of the pressure. This publication already includes a warning on the possible bias caused by the experimental technique, which should be evaluated with caution. In the abovementioned article, another paper by Cass and Hinman [2] was cited that showed no significant influence on urethral pressure by the blockade of striated muscle activity with succinylcholine. In addition, Rud [1] also referred to a paper of Donker [3], who found that even when the striated external urethral musculature was paralyzed, the Urethral Pressure Profile (UPP) only showed a small change. Donker also described a recording of a woman with a normal UPP but without a measurable EMG signal, implying there was no activity of the external striated sphincter. The highest pressure was found in the mid-urethra. Rud also referred to Tanagho [4] who stated that in dogs the urethral striated muscles are responsible for more than 50% of the maximum urethral pressure. However, Tanagho as well as Koff [5] based this assertion on the urethral resistance to urinary flow, which is different from the UPP during rest. 

## 2. Method

The abovementioned reports contain conflicting data that prevent us from drawing a clear conclusion about the elements that contribute to urethral pressure. Therefore, based on the publication of Rud, the work of Donker and the distribution of striated muscle over the urethra, we critically analyzed the existing literature on Pubmed and from other historic sources on the anatomy, position and innervation of the urethral musculature. This was carried out with the intention to find supporting evidence of the contribution of striated and smooth musculature to the maximal urethral pressure during rest and during normal bladder filling.

## 3. Discussion

### 3.1. Anatomy of Urethral Muscles

Huisman [6] depicted the different muscle layers in the mid-urethra on a transverse section. On the outside of the urethra, a horseshoe-shaped striated muscle is located mainly dorsally at the cystoscopic 12 o’clock position and absent ventrally at the 6 o’clock position. The distal part of this muscle is called the compressor urethra (CU), which is located even more distally from the urethro-vaginal sphincter (UVS) [7]. More recently, the anatomy of the striated sphincter has been described by Wallner [8]. Below this striated muscle layer, a circularly oriented smooth muscle layer can be revealed. Between this layer and the mucosa, a longitudinal smooth muscle layer is found.

### 3.2. The Innervations of the Urethral Musculature

The innervation of the urethral musculature is well described by Yoshimura [9]. The striated external muscle is innervated by the pudendal nerve (S2–S4). The smooth (longitudinal and circular) muscle layers both generate spontaneous activity [10] providing the basic tone of the urethra. The smooth circular layer is innervated by the hypogastric nerve (Th 11-L2) and by the pelvic nerve (S2–S4). The hypogastric nerve activates the circular smooth muscle to produce tension above the spontaneous baseline tension. Pelvic nerve activity may relax the circular smooth muscles completely, a process mediated by the production of nitric oxide (NO). The longitudinally orientated smooth muscles are innervated by the pelvic nerve and by the hypogastric nerve. The sympathetic contractile response dominates in the circular layer but is also significant in the longitudinally orientated muscle fibers during bladder filling [11]. This can be explained by increased smooth muscle tone development [12] combined with the most common sympathetic responses [13] in the middle part of the urethra. Moreover, the circularly oriented muscles in this high-pressure zone have been described to generate the highest spontaneous tone [14]. From these animal studies, it is clear that the maximal urethral pressure is determined by smooth muscle activity. This maximal pressure in the mid-urethra, as a result of smooth muscle activity, is corroborated by the findings of Donker. This paper describes the case of a woman with no striated muscle activity at all, which was confirmed by an absent striated muscle EMG signal, but with a normal UPP, with the highest amplitude in the mid-urethra. Both circular and the longitudinal smooth muscle participate in the closure of the urethra during rest and during bladder filling [15]. Donker as well as McGuire [16,17] demonstrated the importance of the sympathetic innervation of the urethra for urethral closure and continence. The external striated muscle or sphincter (EUS) is most likely not involved in creating any urethral pressure during rest and bladder filling. During rest, the MUP is determined as stated above by the activity of the smooth urethral musculature. The main portion of the EUS is situated in the distal part of the urethra and is referred to as UVS and CU. If the EUS can determine the MUP, we should find, based on the anatomy, the highest pressure area in the distal urethra and not the mid-urethra. This hypothesis was confirmed by Kenton [18], who did not find any increase in urethral pressure during bladder filling in 100 participants, despite filling-induced EUS activity increase, which was confirmed by an increase in EMG activity. The EUS function is related to two important mechanisms: the first is that of suppressing bladder contractility by increasing EUS activity [19]. This is an important phenomenon for keeping the bladder relaxed during filling. Most likely, this is the basis of the working mechanisms of several forms of neuromodulations. The second is that under conditions of physical stress, such as a sudden increase in abdominal pressure, an increase in the activity of the EUS can be elicited, including urethral pressure increases due to a reflex contraction of the external striated muscle [20,21]. The EUS has been shown to exert an effect on urethral pressure under circumstances of physical stress, as shown by Constantinou [20]. He observed, during a condition of physical stress, an elevation in urethral pressure in the distal urethra where the EUS is located. The same findings were described by Ulmsten [22]. He described the highest pressure due to the contraction of the EUS in what he calls the ‘knee’, which is at 2/3 of the length from the bladder neck of the pressure profile. This pressure increase at the ‘knee’ of the urethra is only possible by the fixation of the mid-urethra by the pubo-urethral ligaments [23,24]. Kefer [23] found in a rat model that the contraction of the EUS did not elevate the (distal) urethral pressure when the pubo-urethral ligament was deficient, which mimics the transection of the pudendal nerve. This implies that the activity of the external striated muscle itself is not able to increase urethral pressure. The effect of contraction of the external urethral striated muscle under conditions of stress is only seen in the distal urethra. In the EUS area, the urethral pressure can, under these circumstances, become even higher than the MUP, but this pressure increase occurs mainly in the distal urethra and does not add to the MUP. In order to increase the (smooth-muscle-generated) pressure in the mid-urethra, the EUS-evoked pressure must be redirected from its distal position towards the mid-urethra. Considering the physics of the pressures inside the lumen, this transfer is only possible when a fluid column is present in the lumen, which allows an increase in the pressure in the fluid column. Only in this case can the pressure distally generated by the striated muscles be directly added to the pressure generated by the smooth musculature in the mid-urethra. This also explains the findings of Tanagho [4] and Koff [5], as they measured urethral resistance to flow, and thus a fluid column was present in the urethral lumen during these measurements. 

From the arguments mentioned above, there seems to be no pressure contribution of the striated urethral muscle during rest and bladder filling to the MUP. Another disputable interpretation is made with regard to the influence of the urethral mucosa with its venous vascular bed on the urethral pressure. It has been mentioned [1] that the (venous) vascular bed is responsible for 30% of the intraurethral pressure. This assertion is questionable because it is well-known that the artificial occlusion of the vessels in the urethral wall not only decreases the influence of the mucosa on the vascular bed, but also causes hypoxia. Hypoxia decreases the ability of the urethral musculature to maintain a tonic response to alpha adrenergic stimulation [25] and induces smooth muscle relaxation [26]. Hence, hypoxia affects not only mucosal functioning but also the function of the smooth urethral musculature in a negative way.

## 4. Conclusions

The MUP is situated in the medial part of the urethra, a location where the most smooth urethral musculature activity can be seen. The presence and activity of the striated EUS is mainly situated at the distal part of the urethra. During the contraction of the EUS, as seen in situations of physical stress, the main urethral pressure increase can be found in the distal area of the urethra.

Considering the physics of pressure inside a lumen, the redirection of the pressure generated by the EUS to the mid-urethra can only be possible when a fluid column is available in the lumen, which is not the case during rest. The assertion that the MUP is the result of different pressure forces around the urethra, including that of the external striated sphincter, is not supported by evidence and can therefore be considered a flaw in the field of functional urology.

The clinical implication of our finding is that, in patients with urgency incontinence, we should focus not only on bladder function but also consider mid-urethral smooth muscle relaxation, which leads to a complete loss of closure pressure when urine enters the urethra, with this being the cause of complaints by the patient. We know this phenomenon exists, since Schraffordt et al. already showed that mid-urethral tapes cure OAB complaints in about 50% of cases, most likely based on the aforementioned phenomenon [27].

## Data Availability

For this manuscript, no new data were created.

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
