# Peer review of "The Maximal Urethral Pressure at Rest and during Normal Bladder Filling Is Only Determined by the Activity of the Urethral Smooth Musculature in the Female"

_jcm, 2023, doi:10.3390/jcm12072575_

Round 1

Reviewer 1 Report

1.     Please review the article on the anatomy site-specific defect within the urethral stabilizing mechanism published by Ostrzenski A. New site-specific anatomical defects discovered within the female urogenital tract. Eur J Obstet Gynecol Reprod Biol. 2019 Jun;237:164-169. doi: 10.1016/j.ejogrb.2019.04.019. Epub 2019 Apr 23. PMID: 31055155.

2.     Please analyze how your opinion fits the concept of the urethral stabilizing mechanism.

Author Response

We thank the reviewer for the constructive remarks. In analysing the above mentioned article the authors state that their weakness was that they found certain defects but were unable to correlate it with urinary stress incontinence history (page 168) so they were unable to establish the USM site specific defects  with urinary stress incontinence. Beside that the defect was only found in a minority of the cadavers (26.6%). The major urethral stabilising factor are to our opinion the pubo-urethral ligaments. Loss of this stabilising factor is an important cause of urinary stress incontinence because kinking of the urethra is important for closing the urethra under conditions of stress (see Eur Urol 2009;55:932-944).The contraction force of the external striated muscles however is not influenced by the stabilising mechanism of the urethra but, as mentioned above by the authors, the stabilising factor has an influence on the continence during periods of stress. Therefore we did not use this reference in our manuscript that is written under the subchapter ‘opinion’ which definitely is the case here as we consider it an argumented opinion.

Reviewer 2 Report

In the current paper, Authors aims to determine the entities that define the maximal urethral pressure (MUP) in rest and during bladder filling to provide continence in females. They argue that the urethral pressure reflects the urethral smooth muscle activity only. 

Authors should be commended for their efforts; however, I only have some issues/edit suggestions that I would like to point out:

1.     The Authors' point of view is interesting and well-argued but it’s not clear what could be the clinical implication of such observation. Authors should analyze the possible diagnostic-therapeutic impact and its practical usefulness in functional urology in a dedicated paragraph. 

2.     To persuade the readers about their position authors should clarify the studies selection criteria and the historical sources used. 

Author Response

We thank the reviewer for the suggestions and reviewing.  We added a remark on the clinical implication of the finding that the urethral pressure is determined only by the smooth muscle activity. The smooth muscles of the urethra  can be completely relaxed by pelvic nerve activity during bladder filling by NO, with complete loss of urethral closure force, and in these situations giving rise to a strong feeling of urgency or even incontinence without causing a bladder contraction. In many cases of urgency and urge incontinence where we are unable to find an unstable bladder we have realize that an complete smooth muscle urethral relaxation could be the cause of the symptoms. Some lines about clinical implications and a reference are added in the discussion.

Concerning the second item about the study selection: there was no study selection we just tried to find publications that confirm our hypothesis that the MUP is only determined by the smooth muscles base on the findings of Rud, Donker and the distribution of the striated muscle over the urethra. Our manuscript that written under the subchapter ‘opinion’ which definitely is the case here as we consider it an argumented opinion.